# Realizing Innate Potential: CAR-NK Cell Therapies for Acute Myeloid Leukemia

**DOI:** 10.3390/cancers13071568

**Published:** 2021-03-29

**Authors:** Mark Gurney, Michael O’Dwyer

**Affiliations:** 1Apoptosis Research Center, National University of Ireland Galway, H91 TK33 Galway, Ireland; m.gurney1@nuigalway.ie; 2ONK Therapeutics Ltd., H91 V6KV Galway, Ireland

**Keywords:** CAR-NK, acute myeloid leukemia, immunotherapy

## Abstract

**Simple Summary:**

Infusions of T-cells genetically modified to recognize the protein CD19 (CD19 CAR-T cells) have proven a potent form of cancer therapy for certain cancers arising from B-cells. These treatments, while revolutionary, remain expensive to manufacture using a patients’ own cells and can have considerable side effects. There is great interest in improving upon and expanding the reach of these new treatments to other cancer types. Natural killer (NK) cells are an alternative cell population with unique properties which can also be modified to recognize specific proteins (CAR-NK cells). The properties of CAR-NK cells should allow manufacturing from healthy donor cells with rapid availability and potentially fewer side effects. NK cells have an innate ability to target acute myeloid leukemia (AML). In this review article, we consider the potential that CAR-NK cells possess to enhance this effect and offer a new type of immunotherapy for AML.

**Abstract:**

Next-generation cellular immunotherapies seek to improve the safety and efficacy of approved CD19 chimeric antigen receptor (CAR) T-cell products or apply their principles across a growing list of targets and diseases. Supported by promising early clinical experiences, CAR modified natural killer (CAR-NK) cell therapies represent a complementary and potentially off-the-shelf, allogeneic solution. While acute myeloid leukemia (AML) represents an intuitive disease in which to investigate CAR based immunotherapies, key biological differences to B-cell malignancies have complicated progress to date. As CAR-T cell trials treating AML are growing in number, several CAR-NK cell approaches are also in development. In this review we explore why CAR-NK cell therapies may be particularly suited to the treatment of AML. First, we examine the established role NK cells play in AML biology and the existing anti-leukemic activity of NK cell adoptive transfer. Next, we appraise potential AML target antigens and consider common and unique challenges posed relative to treating B-cell malignancies. We summarize the current landscape of CAR-NK development in AML, and potential targets to augment CAR-NK cell therapies pharmacologically and through genetic engineering. Finally, we consider the broader landscape of competing immunotherapeutic approaches to AML treatment. In doing so we evaluate the innate potential, status and remaining barriers for CAR-NK based AML immunotherapy.

## 1. Introduction

Acute myeloid leukemia (AML), the most common acute leukemia in adults, is a biologically heterogenous disease [1]. Genetic alterations driving proliferation or disrupting differentiation lead to the accumulation of immature blast cells, resulting in impaired bone marrow function [2]. Standard anthracycline and cytarabine based chemotherapy is curative for a minority of patients, generally with low-risk disease identified by genetic, clinical and treatment response factors. Sensitive measurable residual disease (MRD) techniques often detect blast cells after treatment despite morphological remission and are increasingly applied to prognostication and treatment planning [3]. Consolidation therapy with allogeneic stem cell transplant (ASCT) may be pursued to reduce the risk of disease relapse balancing patient fitness, donor availability and transplant associated toxicity [4]. Immune mediated graft versus leukemia (GVL) activity, fundamental to the benefit of ASCT, comes at the expense of graft-versus-host disease (GVHD) risk. Relapse, due to the persistence of distinct self-renewing leukemia stem cell (LSC) populations carries a poor prognosis [5]. Several molecularly targeted therapies for specific disease subtypes are now available, providing welcome yet incremental gains [6,7,8,9]. For patients diagnosed in older age, where the disease incidence is highest, available treatments are frequently either unsuitable or unsuccessful. Building upon the GVL activity of ASCT and recent progress in restoring or redirecting immune responses to successfully treat other malignancies, effective and tolerable immunotherapy could revolutionize AML treatment. 

CD19 chimeric antigen receptor (CAR) T-cell therapies are effective treatments for a subset of B-cell malignancies [10,11,12]. The regulatory approval of tisagenleleucel and axicabtagene ciloleucel was the culmination of decades of research, built upon foundational approaches to adoptive cell transfer (ACT) immunotherapy and was enabled by safe and effective genetic engineering techniques [13]. The CAR-T cell field is evolving rapidly with common themes of improving the efficacy and safety of existing therapies and applying their principles to other diseases [14]. While CD19 is an almost ideal tumor associated antigen (TAA), the approval of B-cell maturation antigen (BCMA) CAR-T cell therapies for multiple myeloma confirms that these principles can be transferable [15]. Licensed CAR T-cell products have well defined limitations; adverse effects including cytokine release syndrome (CRS) and immune effector cell-associated neurotoxicity syndrome (ICANS), a complex and expensive manufacturing process from autologous T-cells (to avoid GVHD) and incomplete efficacy [13]. AML is an intuitive, albeit challenging disease to treat with CAR-T cell approaches [16]. Several solutions are under investigation, although in the absence of an ideal target antigen it remains unclear if a viable and widely applicable AML CAR-T cell therapy will emerge.

Natural killer (NK) cells, a lymphocyte population primarily involved in viral and cancer immunity, can also be engineered to express CARs, creating a product with similar principles but distinct characteristics to CAR-T cells (Table 1) [17,18]. NK cells do not mediate GVHD, supporting allogeneic and potentially off-the-shelf application. Innate, human leukocyte antigen (HLA) and antigen independent, target cell recognition provides a separate mechanism of tumor targeting, while early clinical evidence suggests a lower incidence of CRS and ICANS [17]. These factors would combine to reduce the cost of both manufacturing and associated clinical monitoring. NK cells also have a well characterized role in AML biology, contribute to the GVL effect of ASCT and display substantial anti-leukaemic potential deployed as ACT in AML [19,20,21]. The attributes of the CAR-NK platform are especially suited to AML treatment potentially overcoming specific barriers encountered with CAR-T and existing NK ACT. In this article we explore further the potential for CAR-NK based AML immunotherapy. 

## 2. CAR-T Cell Therapy and AML

### 2.1. The Contrast to CD19 CAR-T

The principles of CAR therapies have been extensively reviewed [13,27]. CARs are designed to combine an antigen recognition domain with hinge, transmembrane, and intracellular stimulatory and co-stimulatory domains conferring antigen specific reactivity to an immune cell. ‘2nd generation’ CAR designs combine a T-cell receptor like signaling domain (usually CD3ζ) and a co-stimulatory domain (typically CD28 or 4-1BB) within the same construct, enabling in vivo expansion and persistence, determinants of clinical efficacy [28]. Two CAR-T cell therapies targeting CD19 in pediatric B-cell acute lymphoblastic leukemia (B-ALL) and diffuse large B-cell lymphoma (DLBCL) were approved in the United States in 2017, and the European Union in 2018. Manufacturing of patient-specific products begins when treatment is indicated, using autologous T-cells modified by retro- or lenti- viral transduction during ex vivo expansion at centralized facilities. The products are cryopreserved for transport and administered after lymphodepleting (LD) chemotherapy which creates a niche and supports in vivo expansion. CD19 CAR-T cells achieve durable remissions for some patients, often in clinical scenarios without effective alternatives. Relapse is encountered in both B-ALL and DLBCL broadly due to antigen escape or loss of CAR-T persistence [29]. Investigational CAR-T therapies may solve some of the limitations of established products. Targeted genome editing may allow for safe allogeneic CAR-T cells simplifying the chain of manufacture, while dual targeting and modifications to CAR-T composition could improve efficacy while reducing CRS and ICANs [23,30,31,32]. Non-viral, transposon based CAR-T engineering is feasible and potentially cost-saving relative to viral approaches [25,33]. A vast array of CAR therapies across diseases and target antigens are under investigation [14].

Fundamental to the success of CD19 CAR-T is homogenous CD19 expression on malignant B-cells with restricted expression on normal cells, making CD19 an almost ideal TAA [34]. On-target off-tumor effects of CD19 CAR-T cell therapy are limited to normal B-cells and the resulting hypogammaglobulinemia is manageable. The foremost reason that CAR T-cell therapies have not been readily adapted to AML is the absence of a similarly suitable target antigen [35,36]. Many candidate AML antigens are widely expressed among myeloid cells and off-tumor effects on normal myelopoiesis can be profound. Targeting antigens present on committed myeloid precursors and mature myeloid cells leads to myelosuppression with inherent risks of infection and a requirement for advanced supportive care. If antigens expressed on hematopoietic stem cells (HSC) are targeted, marrow aplasia may result, requiring stem cell transplantation for marrow recovery [37]. A requirement for ASCT rescue will exclude many patients and carries inherent risks for those who are eligible. In addition, a large sink of off-tumor antigen expression could both reduce efficacy and exaggerate adverse effects- disease burden is associated with CRS incidence when targeting CD19 [38]. Beyond off-tumor effects, the heterogeneity of AML within and between patients presents a barrier to the success of CAR-T approaches targeting a single antigen. Furthermore, LSCs are rare within the diseased marrow, have distinct and varying surface immunophenotypes and their targeting is a separate but highly desirable component of AML immunotherapy [39]. While the landscape of LSC heterogeneity has been recently reviewed, prominent therapeutically relevant LSC associated antigens are summarized in Figure 1 [40].

### 2.2. Target Antigens in AML

Despite the absence of an ideal target, pre-clinical and several clinical reports exist of AML CAR-T therapies. CD33 is a well-established target, expressed in almost all AML cases, but exemplifies the challenges discussed above. Antigen negative sub-populations are ubiquitous and expression on myeloid precursors (and possibly HSCs) is challenging [41,42]. Myelosuppression is encountered with the CD33 antibody drug conjugates (ADC) gemtuzumab ozogomycin and vadastuximab talirine and also with CD33 CAR-T cells [43,44,45]. Transient partial response was documented in a clinical case report, although indications of intact hematopoiesis may reflect a low CAR binding affinity [46]. CD123 is also widely expressed in AML and considered a more consistent LSC marker than CD33. Differing CAR designs interacting with low CD123 expression on HSCs may explain the varying myeloablative potential of CD123 CAR-T cells reported [47,48]. Budde et al. described promising findings from 6 patients treated with CD123 CAR-T, including one complete response (CR). Although all participants were required to have an ASCT donor, myelosuppression without myeloablation was documented [49]. Notably, half of the enrolled patients did not receive the planned treatment due to fatal infection, disease progression or failures in leukapheresis and manufacturing [50]. C-type lectin-like molecule-1 (CLL-1, CLEC12A) has similar expression characteristics to CD123, but with less concern for myeloablation reflecting its absence on HSCs. CLL-1 CAR-T has been proposed as a consolidation strategy for AML in remission, as CLL-1 positive blast cells are relatively chemotherapy resistant and this approach reduced relapse in a xenograft model [51]. The difucosylated carbohydrate antigen Lewis-Y was the target of the first clinical AML CAR-T trial conducted, where persistence of CAR-T cells without definite responses was documented and attributed to the low surface density of the antigen [52]. Aberrantly expressed in AML, CD7 CAR-T is also in development, trading myelotoxicity for T-cell depletion and relevant for 30% of AML cases [53].

Other approaches aim to adapt CAR-T to the antigenic landscape of AML. Targeting multiple antigens simultaneously may be fundamental to avoiding relapses from pre-existing antigen negative sub-clones. Perna et al. compared proteomic and transcriptomic data from AML and normal tissues identifying antigen pairs to increase blast targeting without increasing toxicity to normal tissues [36]. Additionally, immunophenotyping of a large cohort of AML cases identified antigen combinations of T-cell immunoglobulin and mucin-domain containing-3 (TIM-3) with either CLL-1 or CD33 as warranting further study [35]. CAR technologies which restrict activation to cells expressing both antigens would restrict off tumor targeting to monocytes yet identify up to 75% of LSCs. A split, universal and programmable (SUPRA) CAR system with vast flexibility, enhanced precision and controllable dosing has been described in which the CAR scFv is replaced by an adaptor protein which then interacts with separate scFv containing constructs [54]. The ability of this system to sequentially target multiple antigens without re-engineering T-cells cells ex vivo, and readily apply complex logic-based approaches to antigen combinations could be especially useful in the setting of AML. Indeed, the pre-clinical development and potential of a modular synthetic agonistic receptor T-cell platform using a similar principle and targeting CD33 and CD123 in AML has recently been described [55]. This versatility will likely come at the cost of a requirement for repeated infusions reflecting the shorter half-life of a separately delivered scFv construct.

While the prospectively identified combinations and novel split CAR approaches described above remain to be tested clinically there have been dual targeting approaches reported. Updated trial results of a dual CD33/CLL-1 CAR-T product which recognizes cells positive for either antigen were recently presented [56]. CAR-T expansion and complete MRD negative remissions were achieved in 7/9 treated patients, with expected marrow suppression, permitting six patients to proceed to reduced intensity conditioning ASCT. An alternative approach to targeting multiple antigens uses the NKG2D receptor as a CAR binding domain. This activating receptor, present on T-cells and NK cells, recognizes 8 NKG2D ligands commonly expressed on AML blasts but not normal cell populations. NKG2D CAR-T cells have now entered clinical trials. Short term responses and poor persistence occur without LD chemotherapy, although successful bridging to ASCT was reported [57,58]. Ongoing trials have been modified to include LD chemotherapy and a modified manufacturing technique supporting an early memory T-cell phenotype [59].

A further alternative to eliminating off-target effects is to target neoantigen peptides presented at the cell surface in an HLA dependent fashion. This is the principle behind a promising HLA-A2 dependent, mutated nucleophosmin 1 (NPM1c) targeted CAR, applicable to 35% of AML cases [60]. Others have sought to reduce expected off-tumor effects associated with target antigens. CRISPR/Cas9 editing has been deployed to knockout CD33 in HSCs, allowing CD33 negative (and CAR-T resistant) hematopoiesis post SCT in animal models [61]. Transient CAR expression through mRNA delivery was evaluated in a phase I clinical trial reflecting concern for persistent off target effects using a CD123 CAR but also appeared to limit efficacy [62]. Finally, modulation of CAR binding affinity allows for tumor specific targeting of antigens with overlapping but increased expression relative to normal tissues. This principle has recently been applied to CD38, an established target in multiple myeloma but also relevant for a proportion of AML cases [63,64]. Despite these innovative and varied approaches, CAR-T therapies for AML remain investigational. For a detailed review of CAR-T in AML readers are referred to reference [16].

## 3. Natural Killer Cells and AML

### 3.1. NK Cell Functions

Natural killer cells are innate lymphocytes which express a collection of germline encoded activating and inhibitory receptors [65]. The balance of signaling through these receptors determines the response of the cell, varying from tolerance (desired for most self-cell interactions), to triggering of a degranulation (perforin and granzyme) response [66]. Target cell death can also be triggered through death receptors via NK cell expression of TNF-related apoptosis-inducing ligand (TRAIL) and Fas ligand [67]. NK cells modulate broader immune responses through release of TNF-α and IFN-γ, the latter of which provides an important stimulus of dendritic cell maturation and thus adaptive immunity [68]. NK cell TRAIL expression also contributes to immunomodulation through death receptor mediated elimination of specific T-cell and dendritic cell subsets [69,70]. Important inhibitory signals to NK cells are delivered by inhibitory killer immunoglobulin-like receptors (KIR) and NKG2A, interacting with their ligands: epitopes present on specific subsets of MHC class I molecules. This system provides for self-tolerance and loss of HLA class I expression on virally infected and transformed cells is the basis of ‘missing self’ NK cell recognition. Although inhibiting mature NK cell responses, intact signaling through inhibitory KIRs and NKG2A during maturation is integral to the process by which NK cells ultimately acquire and calibrate their functional capacity, termed NK cell “licensing” or “education” [71]. Individuals frequently possess a population of hypofunctional NK cells expressing an inhibitory KIR for which the respective ligand is not present in their complement of self HLA molecules. Recruitment of this hypofunctional population by a strong activating stimulus has been recognized [72]. 

In practice, loss of inhibitory stimuli alone is generally insufficient to trigger NK cell cytolytic activity [66]. Activating receptors, such as natural cytotoxicity receptors (NCRs) NKp30, NKp44, NKp46 and the NKG2D receptor, contribute to NK cell activation in response to ligands induced with cellular stress on infected, damaged, or transformed cells. The strong activation signal provided by the FCγRIII (CD16) receptor binding to Fc portions of IgG antibodies on target cells overcomes intact inhibitory signals [73]. This NK cell mediated antibody dependent cellular cytotoxicity (ADCC) contributes an important component of the clinical responses observed to anti-tumor monoclonal antibody therapies. NK cells are phenotypically and functionally heterogenous in vivo. A subset of therapeutic interest are adaptive NK cells which exhibit memory-like properties—enhanced responsiveness after an initial time-limited stimulus—a property which can be induced by combination cytokine exposure ex vivo to create ‘cytokine induced memory-like’ (CIML) NK cells [74]. The unique characteristics of NK cells underly a role in a variety of homeostatic and disease processes including cancer immune surveillance and their growing application to cancer immunotherapy [75].

### 3.2. NK Cells in AML Immunoediting

NK cells are activated by contact to AML blasts in vitro to varying degrees. If NK cell recognition and lysis toward AML in vivo were complete, the disease would not develop. It is evident that intact NK cell activity during disease development not only selects for AML blasts relatively resistant to NK cell targeting, but also processes capable of suppressing NK cell function. As a result NK cells isolated from patients with AML are functionally deficient relative to healthy persons and this is partially rectified for patients achieving complete remission with standard therapy [76]. The quality of subsequent NK cell immune surveillance, reflecting both restoration of NK cell function and relative blast cell expression of ligands for activating and inhibitory NK cell receptors, is implicated in remission maintenance [21,77]. Varied mechanisms underlying escape from NK cell activity in AML have been described, are summarized in Figure 2 and have recently been reviewed in detail [78]. Adoptively transferred NK and CAR-NK cells are expected to rapidly encounter and potentially be inhibited by these same processes [79]. 

The balance of signaling via traditional NK cell activating and inhibitory pathways is skewed in AML. MHC class I expression is generally maintained providing a baseline inhibitory stimulus to NK cells [80]. NKG2D ligands are frequently down regulated including on LSC populations by a variety of mechanisms including proteolytic cleavage producing soluble NKG2D ligands [77,81,82]. Upregulation of the immunosuppressive cell surface glycoprotein CD200 on AML blasts including LSCs is associated with reduced NK cell function through its receptor CD200R [83,84]. The phenotype of NK cells is also modified. Expression of NCRs is reduced relative to healthy controls in an effect reliant on NK and blast cell contact [85]. NKG2D downregulation occurs secondary to chronic exposure to hypoxia, cell surface and soluble NKG2D ligands and via blast and stroma derived TGF-β [86,87,88]. NKG2A expression is often increased, at least partially driven by blast cell production of IL-10, while IFN-γ induces expression of the ligand for NKG2A (HLA-E), on blast cells [89,90]. The DNAM1/TIGIT/CD96 axis is also impacted in AML. DNAM1 and TIGIT represent activating and inhibitory NK cell receptors, respectively for the antigen CD155 which can be expressed on AML blasts and MDSCs. The relative expression of the ligands and receptors of this family vary, but alterations have been implicated in suppressing NK cell responsiveness [91,92]. Recently, sialic acid-based ligands for the NK cell inhibitory receptor siglec-7 have been recognized on CD43 (a sialoglycoprotein frequently expressed on AML blasts) [93,94]. 

Additional processes have been identified which suppress NK cell function in AML. Glycogen synthase kinase-3β (GSK3β) is upregulated in NK cells relative to healthy controls, and inhibits function through reduced expression of LFA-1 and impaired immune conjugate formation [95]. Receptor activator of nuclear factor kappa-Β ligand (RANKL, a TNF family protein) expression on blasts confers bidirectional immunosuppressive effects. Reverse signaling into blast cells leads to secretion of immunosuppressive cytokines including IL-6, while forward signaling via receptor activator of nuclear factor kappa-Β (RANK) on NK cells directly inhibits NK responses [96]. Persistent elevations in IL-6 during induction chemotherapy negatively impact prognosis in AML, which may reflect reduced expression of NK cell activating receptors and stabilization of malignant cell PD-L1 expression previously attributed to this cytokine [97,98,99]. Blast cell production of soluble agonists activates the aryl hydrocarbon receptor (AHR) transcription factor in NK cells, driving microRNA29b expression, interfering with NK maturation and function [100,101]. Reactive oxygen species (ROS) formation, driven by monocytic and myelomonocytic AML subtype expression of the NADPH oxidase component gp91^phox^ causes apoptosis of surrounding immune cells [102]. Immune suppressive cell subsets are recruited and restrict NK cell activity. Myeloid derived suppressor cell (MDSC) expansion is facilitated by soluble NKG2D ligands, while regulatory T cell (Treg) expansion is supported by blast cell production of indoleamine 2,3-dioxygenase (IDO) [103,104]. These subsets inhibit NK cell activity through varied mechanisms including expression of membrane bound transforming growth factor beta (TGFβ) and limiting IL-2 bioavailability [105]. MDSC populations upregulate CD155 in response to ROS, impairing traditional NK cell but not adaptive NK cell function reflecting differing TIGIT expression [106]. Finally, hypoxia has been associated with varied inhibitory NK cell effects including reduced activating receptor expression and impaired production of IFN-γ [88,107]. The transcription factor HIF1α is a clinically relevant mediator of hypoxia related NK cell inhibition in cancer [108]. Thus, endogenous NK cells and those introduced by adoptive transfer face a complex network of inhibitory factors which need to be overcome to mediate anti-leukemic efficacy in AML. The relative importance of these mechanisms may vary between patients, between AML sub-clones, and from diagnosis to remission and relapse [77]. 

### 3.3. NK Cells in ASCT

The decision to pursue ASCT as consolidation therapy in AML considers relapse risk predicted from clinical, genetic and response related factors, versus procedural risks impacted by patient fitness and the degree of HLA matching of available donors [4]. A multifaceted, immune mediated GVL effect confers partial protection against disease relapse after ASCT [109]. T-cell reactivity occurs against minor HLA antigens or mismatched major HLA antigens, if present. This reactivity has traditionally been considered the major component of the GVL response and is closely allied with a risk of GVHD [110]. NK cells also contribute to GVL responses in AML in a manner that is independent of GVHD, and to a greater extent than in other diseases treated with ASCT [19]. NK cell GVL effects may reflect their ‘natural cytotoxicity’ or alloreactive recognition. 

The quantity of NK cells in HLA matched stem cell grafts has been correlated with relapse risk in AML, at least partially mediated by mature NK cells with high DNAM-1 expression with a potential to directly lyse CD112 and CD155 positive blast cells irrespective of alloreactivity [111]. Alloreactive recognition of AML blasts by NK cells is most prominent after haploidentical stem cell transplant. In this setting, donor-recipient iKIR-KIR ligand mismatch occurs frequently, NK cells rapidly recover post-transplant and T-cell depletion of stem cell grafts is often performed to reduce GVHD risk. Thus, it is frequent to have ‘licensed’ donor derived NK cells expressing an iKIR for which the respective KIR ligand is missing in the recipient, supporting NK cell activation through ‘missing self’ reactivity. The presence of alloreactive NK cells in haploidentical grafts has been shown to robustly correlate with reduced disease relapse risk [19,112]. However, haploidentical donors are not favored in general due to increased rates of GVHD and graft failure. For HLA matched donors, where iKIR-KIR ligand mismatch is less likely to occur, the presence of activating KIR2DS1 in donors has been implicated in a clinically relevant graft versus leukemia effect, modulated by the HLA-C phenotype of the donor [113,114]. The presence of KIR2DS1 (which recognizes the HLA-C2 KIR ligand) predicts a reduced rate of relapse in matched, or one locus mismatch ASCT, if the donor is *not* homozygous for HLA-C2. This correlates with in vitro observations of tolerance induced by chronic HLA-C2 exposure, which is not observed with C1/C1 and C1/C2 haplotypes. Recruitment of the population of hypofunctional ‘uneducated’ NK cells expressing iKIRs for ligands found in neither donor nor recipient has also been proposed in the inflammatory milieu post ASCT and was also shown to impact the efficacy of histamine dichloride with IL-2 used as consolidation therapy [72,115]. These observations have established the clinical relevance of NK cell alloreactivity in AML and provide an evidence base for donor choice in NK ACT and CAR-NK settings.

### 3.4. NK Cell Adoptive Cell Transfer for AML

Rare but notable successes with early attempts at ACT immunotherapy, and growing recognition of the alloreactive potential of NK cells in ASCT led to the investigation of purified haploidentical NK cell ACT in AML [116]. The development of LD chemotherapy regimens and exogenous cytokine support have been viewed as important components of this therapy [117,118]. These elements were combined in the seminal description of haploidentical, short term activated NK ACT with systemic IL-2 support by Miller et al. in 2005 [20]. Subsequently a variety of approaches have been reported using short term activated or longer term expanded NK cells, from sources including apheresis product, HSC differentiated NK and irradiated NK cell lines. NK ACT has been investigated in two main scenarios outside the setting of ASCT across a range of single arm early phase clinical trials: in relapsed or refractory AML or as consolidation therapy for patients in remission. These accumulated reports of NK ACT offer insight into the factors associated with clinical responses and define the baseline activity on which CAR modified NK cell approaches seek to build.

In their initial description, Miller et al. reported complete responses (CR) in 5/19 patients with relapsed/refractory AML, associated iKIR-KIR ligand mismatch with positive outcomes, and documented NK cell expansion in vivo (supported by intense LD chemotherapy and an associated rise in IL-15) [20]. Emphasizing the importance of the tumor microenvironment (TME) and reflecting concerns that systemic IL-2 can expand Treg populations, the group went on to demonstrate improved persistence and response rates of >50% with prior depletion of Treg cells using a recombinant IL-2-diptheria toxin protein [119]. These impressive responses likely occurred rapidly as the earliest measures of chimerism and Treg depletion were correlative, and the window of IL-15 increase was brief. Improving NK cell persistence was identified as a priority. As an alternative approach to avoiding Treg stimulation the group utilized exogenous rhIL-15 in place of IL-2, demonstrating a 32–40% CR/CRi rate [120]. In this study the relationship between responses and NK cell expansion in vivo was not observed, and CRS/ICANS was noted with subcutaneous, but not intravenous IL-15. Exogenous IL-15 however stimulates CD8+ T-cells introducing a concern for rejection of donor NK cells. Applying a similar LD and IL-2 supported approach in a group of AML patients in morphologic remission not considered eligible for ASCT, Curti et al. demonstrated an association between the percentage of alloreactive NK cells infused and subsequent relapse rates [121]. Bjorklund et al. applied a lower intensity LD regimen and administered haploidentical NK cells without IL-2 support successfully documenting responses in de novo AML and MDS/AML [122]. Rubnitz et al. reported application of a similar approach using systemic IL-2, and a lower intensity LD regimen in a pediatric setting, for intermediate risk AML in remission. The treatment was well tolerated and NK cell expansion occurred, although no benefit in preventing relapse was demonstrated in a small phase II trial [123,124]. CIML NK cells administered after LD chemotherapy and with systemic IL-2 support robustly expand in vivo and also induce remissions in a proportion of patients with relapsed/refractory AML [74]. Pre-clinical in vitro and xenograft models show enhanced anti-leukemic activity of CIML NK versus control NK cells. Irradiated NK-92 cells appear safe when applied as an off the shelf product, although lacked persistence and were of limited efficacy in a phase I trial [125]. As an alternative cell source, Dolstra et al. described the use of NK cells differentiated from cord blood derived CD34 progenitor cells, as a remission maintenance strategy. The approach appears safe but low level chimerism of short duration was observed in the absence of cytokine support [126].

These reports, and others, provide an insight into the factors which influence clinical responses to NK ACT in AML. Haploidentical NK cell ACT achieves complete responses for a subset of patients. Responses are generally short in duration but have provided bridging to potentially curative ASCT in some cases. In contrast to ASCT where alloreactive NK cell populations are replenished from donor derived hematopoiesis, available measures suggest haploidentical NK ACT leads to a limited window of expansion and persistence despite exogenous cytokine support. Notably, LD chemotherapy of varying intensity has been relied upon to support NK cell expansion, one parameter that has been associated with clinical responses. While immune mediated adverse effects have been few, those relating to marrow suppression post LD chemotherapy have been encountered. In certain cases, iKIR-KIR ligand mismatch is associated with responses and modulation of TME elements also appears to influence success. Evidence for immunoediting of residual blast populations has been documented [122]. 

## 4. CAR-NK: A Compelling Platform for AML Immunotherapy?

CAR-NK cells combine the antigen specific targeting of CAR-T cells with the innate activity of NK cell ACT creating a platform which overcomes many of the limitations of each of these therapies applied to AML. In the absence of an ideal CAR target antigen, the well-defined innate and alloreactive potential of NK cells against AML provides for activity against antigen negative subclones. NK cell ADCC in the presence of monoclonal antibodies could provide a simple and time limited dual targeting solution, with mechanistic similarities to the modular CAR designs in development. Several approaches to engineering NK cell persistence have emerged coupled with CAR-NK cell development, which could overcome a consistent limitation of NK ACT. These factors, along with a potential for ‘off-the-shelf’ application, and a safer adverse effect profile define an attractive approach to AML immunotherapy. CAR-NK cell application is also adaptable to both clinical scenarios discussed previously, with overlapping but distinct challenges. In relapsed/refractory disease, activity against bulk AML blasts is required, and maximal TME effects can be anticipated. Alternative therapies, where existing, are also generally myelosuppressive, and bridging to ASCT in this setting is often the established goal— supporting the use of myeloablative antigens. Rapid availability of CAR-NK products would be a distinct benefit here. For AML in remission, CAR-NK would be applied as a consolidation therapy initially for patients not considered candidates for ASCT. In this setting, there is a greater emphasis on LSC targeting, a diminished TME, but also a lower tolerance for persistent myelosuppressive effects beyond LD, as the alternative therapy may be observation or well-defined chemotherapy-based consolidation. Rapid availability is less essential, but tailoring therapy to a detected LSC immunophenotype may be feasible and could be combined with MRD based techniques to further select patients based on relapse risk. The versatility of CAR-NK therapies could provide solutions for each of these clinical scenarios. In this section we will consider the status of CAR-NK cell development for AML and remaining barriers to be overcome. 

### 4.1. Principles of CAR-NK Therapies

CAR-NK continues to represent a small segment of a landscape of CAR therapies dominated by T-cells [14]. Active and completed clinical trials investigating CAR-NK cell therapies are summarized in Table 2, and have also been compiled in detail in recent articles [127,128]. The pioneering phase I clinical trial at MD Anderson Cancer Center of cord blood derived CD19 CAR-NK cells represents the most advanced clinical data available and provides important insights for AML CAR-NK development [17]. Using HLA-mismatched NK cells transduced with a transgene including a CD19 CAR, IL-15 and inducible caspase 9, Liu et al. demonstrated responses (including CR in 7/11 patients) in chronic lymphocytic leukemia and B-cell lymphomas without GVHD, CRS or ICANS. Notably, CAR-NK cells expanded and persisted for at least one year post infusion, perhaps reflecting autocrine IL-15 stimulation. Despite this persistence, CD19 positive relapses occurred without repeat expansion of CAR-NK cells. The authors report that this platform is capable of scaling to produce 100 doses from a single cord unit, which if successfully combined with cryopreservation would create a truly off-the-shelf product, and plan to expand the project across other diseases, including AML [129]. Clinical reports of other CAR-NK therapy formats exist. Transient mRNA based NKG2D CAR expression was applied in primary NK cells for a clinical trial using local delivery to treat metastatic colorectal cancer with tumor responses reported [130]. CAR NK-92 cell data has been reported in a small phase I trial (discussed below), and several other CAR NK-92 trials are ongoing. 

Human induced pluripotent stem cells (iPSCs) can be stably gene modified and subsequently differentiated into CAR-NK cells, creating a platform with several advantages [131]. A master iPSC cell bank can be cryopreserved, harbor multiple gene edits and produce a homogenous final product. The iPSC platform is also compatible with non-viral transposon-based gene delivery systems, which to date have not been readily applied in primary NK cells [26]. While current manufacturing protocols are longer than for other cell therapy products, the development of an ongoing manufacturing pipeline would allow for off-the-shelf application. Similarly to cord blood derived NK cells, iPSC NK cells display an immature phenotype when compared with peripheral blood NK cells, although this does not appear to diminish their functional capacity [132]. A clinical trial using multiplex gene engineered iPSC derived CD19 CAR-NK cells, also expressing an IL-15/IL-15 receptor fusion protein supporting persistence, and a high affinity CD16 receptor to support enhanced ADCC (FT596) in combination with an anti-CD20 monoclonal antibody is underway. Given the versatility of this approach, early indicators of efficacy and safety are eagerly awaited [133].

The effect of T-cell composition on CAR-T products is now well recognized [134,135]. NK cells also display phenotypic and functional diversity with recognizable maturation stages potentially modulating the characteristics of CAR-NK products [136]. While CAR expression has been shown to activate hypofunctional, uneducated and less mature NK cell subsets in vitro, relatively greater CAR-NK functional potential was observed for mature, adaptive and educated populations [137]. Leveraging this effect, CIML NK cells modified to express a CD19 CAR showed synergism of CAR activation and CIML NK features in a model of relapsed lymphoma [138]. Inhibitory NKG2A signaling, most relevant for immature NK cell subsets, does appear to be overcome by CAR mediated activation, however inhibitory KIR interactions were capable of dampening CAR-NK activation [137]. Thus, the exact NK cell subset composition and KIR/HLA interactions warrant careful consideration in the development of CAR-NK products. CARs which are designed to signal with NK rather than T-cell based signaling domains may further tune the activation of CAR-NK cells, and this topic has been recently thoroughly reviewed [18]. 

**Table 2 cancers-13-01568-t002:** Active and completed clinical trials of CAR-NK cell therapies. B-ALL = B-cell acute lymphoblastic leukemia, CLL = chronic lymphocytic leukemia, NHL = non-hodgkin lymphoma, AML = acute myeloid leukemia, MDS = myelodysplastic syndrome, iPSC = induced pluripotent stem cells.

Cell Source	Target	Disease	NCT Identifier	Status [Reports]	Location
Cord Blood	CD19	B-ALL/CLL/NHL	NCT03056339	Active [17]	MD Anderson Cancer Center
Cord Blood	CD19	B-ALL/CLL/NHL	NCT04796675	Active	Huazhong University of Science and Technology
Haplo NK	CD19	Pediatric B-ALL	NCT00995137	Completed	St Jude Children’s Research Hospital
Haplo NK	CD19	B-ALL	NCT01974479	Suspended	National University Hospital Singapore
Haplo NK	NKG2DL	AML/MDS	NCT04623944	Active	Multiple Sites (USA)
PBNK	NKG2DL	Solid Tumors	NCT03415100	Active [130]	Guangzhou Medical University
iPSC-NK	CD19	CLL/NHL	NCT04245722	Active [133]	University of Minnesota Masonic Cancer Center
NK-92	ROBO1	Solid Tumors	NCT03940820	Active	Suzhou Cancer Center
NK-92	ROBO1	Pancreatic Cancer	NCT03941457	Active [139]	Shanghai Ruijin Hospital
NK-92	BCMA	Multiple Myeloma	NCT03940833	Active	Nanjing Medical University
NK-92	HER2	Glioblastoma	NCT03383978	Active	Johann Wolfgang Goethe University Hospital
NK-92	CD33	AML	NCT02944162	Completed [140]	Jiangsu Institute of Hematology

### 4.2. AML Target Antigens and CAR-NK

Despite the potential benefits of an NK cell approach, the pool of AML relevant CAR target antigens remains unchanged. It is also important to consider NK cell expression of potential targets to account for possible fratricide which could limit efficacy. Furthermore, differential sensitivity to NK cell innate and alloreactivity between the antigen negative populations for each target, could influence target choice using a CAR-NK approach. Most reports of CAR-NK cell activity at present describe pre-clinical findings. The status of CAR-NK therapies in development for prominent AML antigens is summarized in Table 3.

CD33 expression has been documented in subsets of activated NK cells (acting as an inhibitory receptor) but sufficient to impair NK viability using a tri-specific immune-ligand molecule [152,153]. NK cell CD33 knockout could thus serve a dual purpose of reducing fratricide and enhancing NK cell activation in a CD33 CAR-NK design. CD33 CAR NK-92 cells have been evaluated in a small phase I clinical trial by Tang et al. [140]. While multiple infusions of CD33 CAR-NK-92 appeared safe following salvage chemotherapy in relapsed disease, it is not possible to infer anti-leukaemic effect in this study and the focus remains overcoming the limitation of irradiation. Recently, targeted gene insertion into a safe-harbor locus via homologous repair using CRISPR/Cas9 gene editing in combination with adeno-associated virus (AAV)-mediated gene delivery was used to generate primary CD33 CAR-NK cells with confirmation of in vitro CD33 positive AML targeting [142]. Several reports covering peripheral blood, cord blood and NK-92 cells expressing a CD123 CAR confirm in vitro activity, although Christodoulou et al. failed to see leukaemic control in a xenograft model [129,143,144,145,154]. The latter observation was associated with poor CAR-NK persistence, highlighting a need for separate measures to enhance persistence beyond CAR expression. Our group and collaborators have recently reported the in vitro activity of affinity optimized CD38 CAR-NK cells in AML [148]. CD38 is expressed on primary expanded NK cells, introducing a risk of NK cell fratricide, which can be overcome by CRISPR/Cas9 CD38 knockdown during manufacturing. Interestingly, CD38 knockdown in NK cells is associated with augmented rather than impaired activity, and resistance to oxidative stress relative to wild type NK cells [155,156]. 

Although NK cells express the NKG2D receptor, expression of an NKG2D CAR enhances NK cell activity and should not be subject to downregulation encountered with endogenous NKG2D in AML [157]. NKG2D CAR NK-92 and primary CAR-NK cells have been evaluated pre-clinically against multiple myeloma, and activated CAR-NK cells appeared to outperform CAR-T cells in one animal model [147,158]. A phase I clinical trial of a haploidentical donor derived CAR-NK product targeting NKG2DL in MDS and AML has recently opened (NCT04623944). An existing clinical report of transient NKG2D CAR expression and local delivery leading to responses in colorectal cancer is notable as other examples which purposefully limit the duration of CAR expression have not proven successful [130]. Local delivery is not feasible in AML but modifying the cells further to enhance homing may optimize the early activity of transiently expressed CARs. Dong et al. recently presented pre-clinical activity of CAR CIML-NK cells targeting NPM1c/HLA-A2 [149]. Relative to NPM1c CAR-T cells, it could be assumed that antigen escape via loss of HLA class I expression would be mitigated due to a concurrent enhancement of innate CIML-NK activity [60]. Interestingly, not all off tumor activity can be considered negative when evaluating target antigens. CD33 and CD38 are expressed on MDSC populations, and NKGD2L may also mark Treg cells for elimination supporting a more NK cell favorable TME [159,160,161]. 

### 4.3. Engineering Persistence: A Balance of Efficacy and Off-Tumor Effects

The question of persistence applied to AML for CAR-T and CAR-NK cell therapies is more complex than for CD19 positive malignancies. The limitations in persistence of some CAR-NK cell designs, could be viewed as beneficial in limiting the duration of myelosuppressive effects for certain antigens. Accumulated experience in NK ACT and CAR-T cell therapies, suggests that this limitation in persistence may also compromise efficacy. The ideal balance remains to be defined, and may vary across target antigens, NK cell sources, and clinical scenarios. This concern does not apply to some targets already discussed (e.g., NKG2D and NPM1c CARs), which do not entail the same risk of off-target effects, where long term CAR-NK persistence can be viewed as being a desirable characteristic. In other cases, and relevant for CAR-NK but not CAR-T applications, uncoupling the persistence of CAR signaling from haploidentical NK cell persistence could provide an ideal balance, given the established role of alloreactive NK cells in AML remission maintenance post ASCT. Optimized application of transient CAR expression, or inducible CAR/co-stimulatory signaling could realize this concept. 

Several means of supporting NK cell persistence have been reported which aim to avoid the need for exogenous non-targeted cytokine support. The MD Anderson cord blood CAR-NK process includes an IL-15 domain which provides for secretion and autocrine stimulation of NK cells without increasing circulating IL-15 levels when applied clinically [17]. FATE therapeutics iPSC CAR-NK product FT596 includes a constituently active IL-15/IL-15 receptor fusion domain to provide autonomous signaling [133]. Several groups have reported on the role of knocking out the cytokine-inducible sh2-containing protein (CISH) gene, a negative regulator of IL-15 signaling, enhancing NK cell metabolism, cytotoxicity, NK cell persistence and CAR-NK functionality [162]. Notably, CISH knockout NK cells mediated improved disease control in an AML xenograft model [163]. Wang et al. recently reported the benefit of inducible Myd88/CD40, providing co-stimulation as a freestanding protein separate to the CAR construct, enhancing persistence and cytotoxicity of CD123 CAR-NK cells. Myd88/CD40 represents a common downstream signaling molecule for cytokines known to support NK cell function [164,165]. Notably, in this study, first generation CAR expression, autocrine IL-15 and Myd88/CD40 expression were required for tumor control in a xenograft model, and the authors suggest that separating these signals could also abrogate exhaustion in the long-term. CIML-NK cells display improved persistence in xenograft models, but the extent to which this translates to clinical applications continues to be explored [74,166].

### 4.4. Next Steps: Optimizing CAR-NK Activity in AML

AML biology and clinical experiences with CAR-T and NK ACT infer a need to consider approaches augmenting CAR-NK cell therapies beyond antigen specificity. Gene transfer technologies, and pharmacological approaches to modulating NK cells, AML blasts and the TME are summarized in Figure 3. We will consider the varied pathways to improving CAR-NK activity through NK cell homing capability, disrupting inhibitory checkpoints, modulating AML ligand expression, dual targeting, and NK cell persistence (previously discussed).

#### 4.4.1. CAR-NK Cell Homing

The density of NK cells on bone marrow core biopsies performed after NK ACT for AML correlates with clinical responses [167]. Rapid homing of NK cells to bone marrow could allow for activity before in vivo suppressive factors act and increase the effective ratio of CAR-NK cells to targets. This aspect of CAR-NK design may be especially relevant for approaches that seek to leverage transient CAR expression to balance off-target effects. Depending on the method, expression of the chemokine receptor CXCR4, as well as that of E-selectin ligands, can decrease during ex vivo NK cell expansion and the gain of function variant CXCR4^R334X^ has been shown to enhance NK cell homing to bone marrow when introduced prior to adoptive transfer in animal models [168]. The tethering of circulating leukocytes to endothelium, the first step in leukocyte transmigration, is predominantly mediated by endothelial E-selectin binding of leukocyte E-selectin ligands (cell surface glycolipids and glycoproteins such as CD44, expressing sialyl Lewis X). Bone marrow vasculature constituently expresses E-selectin but this is upregulated in AML, an important mechanism by which leukemic cells (which also highly express E-selectin ligands) maintain their bone marrow niche and evade chemotherapy [169]. The application of translational glycobiology to CAR-T engineering has recently been reviewed [170]. Sialyl Lewis X expression can be enhanced on NK cells through fucosyltransferase activity, the key enzymatic step which determines sialyl Lewis X production in humans. This principle was shown to enhance bone marrow homing in a xenograft lymphoma model using NK-92 cells treated ex vivo with human Alpha-(1,3)-fucosyltransferase (Fut6) and GDP-fucose [171]. 

#### 4.4.2. Inhibitory Checkpoints

Inhibitory pathways represent potential NK cell checkpoints modulating CAR and innate signaling in NK cells. Genetic engineering could provide for robust knockdown of inhibitory receptors, although caution is warranted given the importance of certain inhibitory pathways in NK cell education and licensing, where a separate and carefully timed antibody-based receptor blockade may be preferable. Monoclonal antibodies to many of these targets have been developed and evaluated in clinical trials across a variety of hematological and solid organ malignancies, and several appear especially relevant to AML therapy [172,173]. Despite promising pre-clinical findings a monoclonal antibody blocking the interaction between inhibitory KIRs (KIR2DL1/2/3) and HLA-C, lirilumab, failed to prolong leukemia free survival for elderly AML patients in remission as monotherapy, and notably triggered concern that continuous KIR inhibition could impair NK cell immune surveillance [174]. The study group proposed that a secondary activating stimulus may be necessary to see benefit from KIR inhibition. *In vitro* data supports the concept that iKIR can dampen CAR signaling, although clinical exploration of this combination would require a careful dosing schedule [137]. 

NK cell NKG2A is upregulated in AML, and a population of strongly NKG2A inhibited NK cells dominates recovery post haploidentical ASCT which has prompted investigation of the NKG2A monoclonal antibody monalizumab in this setting [175,176]. Interestingly, in vitro data suggests CAR expression overcomes NKG2A negative regulation in NK cells, although an enhancement of innate signaling could still be an important contributor to overall efficacy. Elimination of surface NKG2A expression enhanced NK cell activity against primary AML blasts and no indications of NK cell impairment were seen in xenograft solid tumor models [177,178]. AML is a disease with a relatively low mutational burden and poor responses to programmed cell death protein 1 (PD-1) blockade were encountered with monotherapy [179]. Combination approaches and augmenting alloreactive T-cell responses post ASCT are now the focus of investigation [180]. NK-cells and blast cells do express PD-1 and PD-L1, respectively and blockade using a PD-1 monoclonal antibody or scFv enhanced NK cell cytotoxicity against AML targets in vitro, although appears to be most relevant for resting rather than cytokine activated NK cells [181]. Inhibition of TIGIT in AML is intuitive, however conflicting reports regarding its effect in experimental systems in vitro suggest caution is warranted [92,182]. An antibody recognizing CD200 positive blast cells enhances the activity of cytokine induced killer cells, and the expression of CD200 on AML LSCs suggests this may be a useful pathway to target [183,184]. Isolation of NK-92 cells negative for the NK inhibitory receptor Siglec-7 (which recognizes sialic acid containing siglec-ligands on the target cell surface), identified enhanced cytolytic activity against AML blasts -while disruption of the interaction of Siglec-7 with CD43 (a dominant source of Siglec-7 ligands) also enhanced NK cell anti-leukemic activity in vitro [93,185]. Denosumab (a clinically available monoclonal antibody against RANKL) has been shown to overcome the immunosuppressive signaling associated with blast cell RANKL and interactions with NK cell RANK [96]. 

#### 4.4.3. AML Ligand Expression and Dual Targeting

Combining with treatments which modulate the expression of the CAR target antigen, or the NK cell receptor ligand profile of blast cells is another route to enhancing CAR-NK activity. Hypomethylating agents, ATRA and HDAC inhibitors modify NKG2D ligand expression in AML especially relevant to NKG2D CAR therapy but also antigen negative innate reactivity, while the CDK inhibitor dinaciclib enhances NK cell recognition of AML blasts predominantly by reducing blast cell HLA-E expression [186,187,188,189]. Suppression of NKG2D ligands on LSCs specifically can be overcome by poly (ADP-ribose) polymerase 1 (PARP1) inhibition [81] PARP1 inhibition has also been shown to induce death receptor 5 (DR5) on AML blasts, sensitizing to killing by TRAIL [190]. Immunomodulatory agents lenalidomide and pomalidomide enhanced NK cell cytolytic activity against AML via a heterogenous effect on NKG2D ligands and CD155 and a consistent decrease in HLA class I expression [191]. ATRA also upregulates CD38 acting through a retinoic acid response element in the CD38 gene, and sensitizes KG1a cells and primary AML samples to CD38 CAR-NK activity [148]. Recent evidence suggesting a negative effect of direct ATRA exposure on NK cell cytotoxicity warrants caution in the timing of this combination [142].

While dual CAR targeting may increase the proportion of AML blasts eliminated by antigen specific recognition as discussed previously, some mechanisms of immune escape may still persist, including failure of the leukaemic cells to bind perforin and resistance to granzyme mediated cell death through leukemic expression of serpinB9/protease inhibitor-9 [192,193]. Here, an alternative mechanism of killing through death receptor pathways may be required. TRAIL is expressed by NK cells, and is considered a desirable feature of ‘activation’ in the context of NK cell expansion. Recent evidence suggests that death receptor mediated killing may be particularly important after pre-formed stores of perforin and granzyme are depleted, providing a backup mechanism of cytotoxicity [194]. Ongoing ‘serial’ death receptor mediated killing also appears to be limited however, perhaps due to exhaustion of pre-formed death ligand, or degradation within the immune synapse during an initial killing event. NK cells can be engineered to constitutively express novel high affinity TRAIL variants with specificity for one or other TRAIL receptor, reducing the role of ‘decoy’ receptors and presenting a viable approach to enhanced death receptor mediated killing [195]. The expression of these TRAIL in a membrane bound form induces optimal death receptor clustering, propagating a potent apoptotic stimulus [67]. A DR4 variant TRAIL is under investigation to mitigate antigen escape in AML [196]. Pharmacological approaches to modulating TRAIL receptor expression could also be applied—the proteosome inhibitor bortezomib is considered the most well-known, and clinically relevant TRAIL sensitizing agent via upregulation of DR5 expression [197]. An element of caution is also required when engineering NK cell death ligand expression- acknowledging the role that death receptor mediated killing has in modulating immune responses and the potential for fratricide [69,198,199].

## 5. CAR-NK in the Context of AML Immunotherapy

At the essence of CAR-NK cell therapies is the conferral of antigen specific targeting to alloreactive NK cells, augmenting their innate anti-leukaemic activity. The overlapping fields of CAR-T and NK ACT have been considered above, however several other prominent and related approaches to AML immunotherapy are in development. Cytokine administration has been investigated to enhance endogenous immunity against AML. IL-2 in combination with histamine dichloride improved LFS in a phase III trial as a consolidation therapy likely through recruiting NK cells, but has not been widely adopted into clinical practice [72]. The IL-15 superagonist ALT-803 achieved responses in patients with relapse post ASCT, and may be combined with other NK cell directed therapies or ACT in the future [200]. Unconjugated antibodies alone have not proven beneficial in AML. The ADC gemtuzumab ozogomycin improves overall survival when combined with induction chemotherapy in CD33 positive favorable and intermediate risk AML [201]. ADCs against other targets are in development [202]. Bispecific and trispecific antibody constructs engage immune cells and targets through simultaneous recognition of a cancer antigen and immune cell surface proteins. This principle has been explored for NKG2DL targeting using an NKG2D-CD16 fusion protein [203]. Trispecific killer engager (TriKE) constructs include an IL-15 moiety to stimulate as well as recruit NK cells, and are in development targeting CD33 and CLL-1 [204,205]. These novel agents recruit endogenous NK cells in an antigen specific manner. Relative logistical simplicity is a potential advantage over CAR-NK, although whether the inherent dysfunction of NK cells in active AML will be sufficient to constrain this elegant approach remains an active question, and the heterogenous expression of these antigens is equally relevant here. Application of these agents as post remission therapy with restored NK cell function or indeed combination with NK cell adoptive transfer which would also permit alloreactivity may ultimately be considered. T cells with engineered TCRs can target a range of intracellular antigens including both TAAs and neoantigens. This bypasses the heterogeneity in cell surface antigens. The TAA Wilms tumor 1 (WT1) has been investigated as a target of both a high affinity engineered TCR-T cell therapy and a vaccine based approach post ASCT with favorable outcomes [206,207].

## 6. Conclusions

CAR-NK cell therapies are at an early stage of investigation in AML, but present an appealing combination of antigen specific, innate and alloreactive activity supported by engineered persistence. The closely related fields of CAR-T cell therapy and NK cell ACT have provided a wealth of information which can be applied to expedite the development of this platform, although challenges remain. The range of NK cell sources, expansion methods, CAR designs and combination approaches offer a toolbox with which to tackle the inherent complexity of AML treatment. If the potential of CAR-NK is realized, this approach may strike an ideal balance in the challenging but expanding field of AML immunotherapy and ultimately improve treatment responses while reducing toxicities.

## Figures and Tables

**Figure 1 cancers-13-01568-f001:**
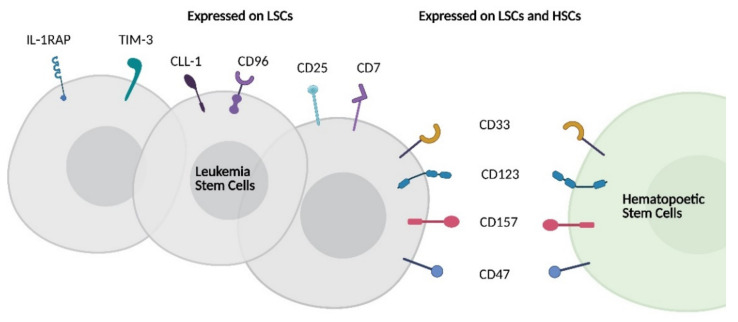
Schematic representing LSC antigen heterogeneity and overlapping expression of therapeutically relevant LSC antigens with HSCs. LSC = leukemia stem cell; HSC = hematopoietic stem cell; IL-1RAP = IL-1 receptor accessory protein, TIM-3 = T-cell immunoglobulin and mucin-domain containing-3; CLL-1 = C-type lectin-like molecule-1.

**Figure 2 cancers-13-01568-f002:**
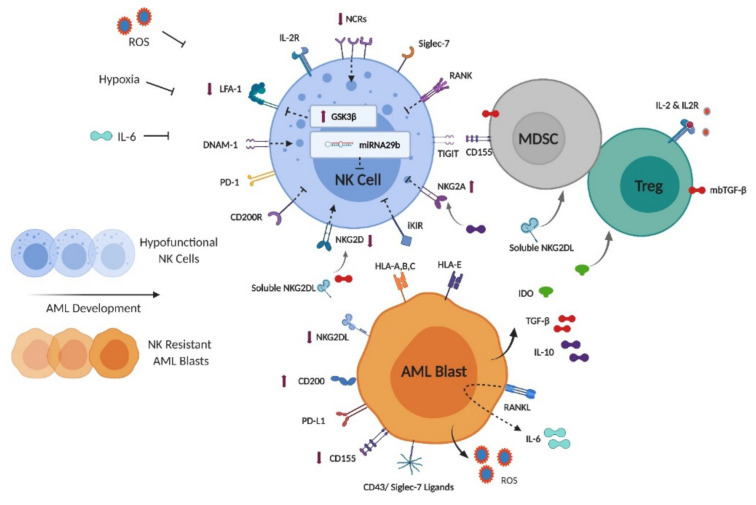
NK cell, blast cell and soluble factors in NK cell-AML immunoediting. ROS = reactive oxygen species; NCR = natural cytotoxicity receptor; LFA-1 = lymphocyte function-associated antigen 1; RANK = receptor activator of nuclear factor kappa-Β; TIGIT = T-cell immunoglobulin and immunoreceptor tyrosine-based inhibitory motif (ITIM) domain; DNAM-1 = DNAX accessory molecule-1; PD-1 = programmed cell death protein 1; iKIR = inhibitory killer immunoglobulin receptor; MDSC = myeloid derived suppressor cell; Treg = regulatory T-cell; mbTGF-β = membrane bound transforming growth factor beta; PD-L1 = programmed death-ligand 1; IDO = indoleamine 2,3-dioxygenase; GSK3β = Glycogen synthase kinase-3β.

**Figure 3 cancers-13-01568-f003:**
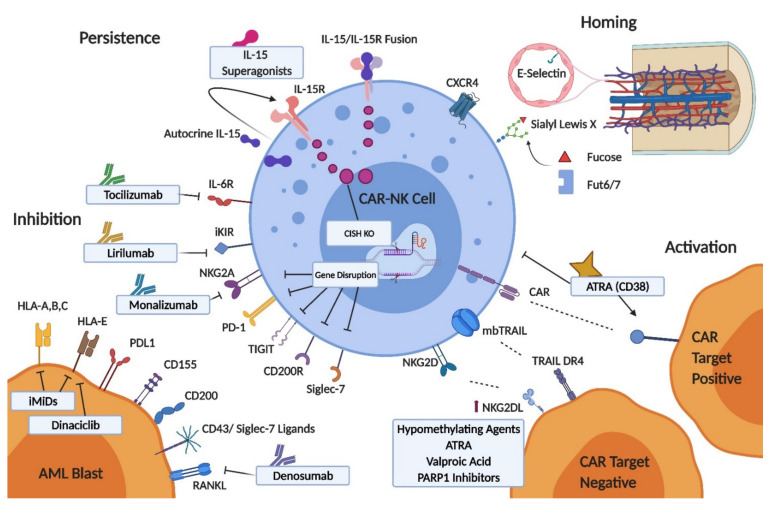
Pathways to augmenting CAR-NK in AML. RANK = receptor activator of nuclear factor kappa-Β; TIGIT = T-cell immunoglobulin and immunoreceptor tyrosine-based inhibitory motif (ITIM) domain; PD-1 = programmed cell death protein 1; iKIR = inhibitory killer immunoglobulin receptor; PD-L1 = programmed death-ligand 1; RANKL = receptor activator of nuclear factor kappa-Β ligand; iMiDs = immunomodulatory drugs; FUT6/7 = alpha-1,3-fucosyltransferases 6 and 7; ATRA = all-trans retinoic acid; PARP1 = Poly (ADP-ribose) polymerase 1; CISH = cytokine-inducible sh2-containing protein; CAR = chimeric antigen receptor.

**Table 1 cancers-13-01568-t001:** Comparing the characteristics of CAR-NK to autologous and allogeneic CAR-T therapies. iPSC = induced pluripotent stem cell, CRS = cytokine release syndrome, ICANS = immune effector cell-associated neurotoxicity syndrome.

	Autologous CAR-T	Allogeneic CAR-T	CAR-NK
Efficacy	Established [11,12,15]	Investigational [22,23]	Investigational [17]
Allogeneic Sources	NA	Yes	Yes
Mechanism of Activation	CAR	CAR	CAR and Innate
CRS/ICANS	Established	Anticipated	Likely Reduced [17]
Cost of Product	$370,000–475,000	Likely Reduced	Likely Reduced
Cost of Care	Variable	Likely Equivalent	Potentially Reduced
Cryopreservation	Established	Established	Investigational
Viral Gene Delivery	Established, Feasible	Established, Feasible	Lower Efficiency
Non-Viral (Stable) Gene Delivery	Clinical Trials [24,25]		Described in iPSCs [26]

**Table 3 cancers-13-01568-t003:** Expression patterns and CAR-NK development status for prominent AML target antigens. CRC = colorectal cancer.

Target	Cases *	HSC *	LSC *	NK	CAR-NK Development
CD33	88% [141]	Yes	Yes	Yes	Clinical: CAR NK92 (NCT02944162) [140] Preclinical: Primary CD33 CAR-NK [142]
CD123	78% [141]	Yes	Yes	No	Preclinical: Primary CAR-NK and CAR NK-92 [143,144,145]
CLL-1	77% [146]	No	Yes	No	
NKG2DL	70% [87]	No	No	No	Clinical: AML (NCT04623944), CRC (NCT03415100) [130] Preclinical: Activity in Multiple Myeloma [147]
CD38	70% [64]	No	No	Yes	Preclinical: PB CAR-NK and CAR-KHYG-1 [148]
NPM1c	35% [60]	No	Yes	No	Preclinical development of CIML-NK CAR [149]
CD7	30% [150]	No	Yes	Yes	Preclinical: CD7 CAR-NK92 in T-ALL [151]
TIM-3	87% [35]	No	Yes	Yes	
CD96	51% [39]	No	Yes	Yes	

* Varying reports likely reflect heterogeneity and a spectrum of positivity. A simplified interpretation is provided.

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
