# Peer review of "Realizing Innate Potential: CAR-NK Cell Therapies for Acute Myeloid Leukemia"

_cancers, 2021, doi:10.3390/cancers13071568_

Round 1

Reviewer 1 Report

Please offer Figures with a better resolution.

Please make a Table with the current clinical trials using CAR-NK cells in AML and other diseases.

Overall, the review is well made and I would find it suitable for publication.

Author Response

Comment 1: Please offer Figures with a better resolution.

Response 1: We appreciate this feedback – the included images in the revised manuscript are at a higher resolution than the initial submission and meet the requirements of the journal.

Comment 2: Please make a Table with the current clinical trials using CAR-NK cells in AML and other diseases.

Response 2: We thank the reviewer for this suggestion, and have added this table (Table 2 in the revised manuscript). We have ensured to reference recent articles which provided similar, and detailed accounts of active clinical phase CAR-NK studies.

Overall, the review is well made and I would find it suitable for publication.

Reviewer 2 Report

The review is very well written and covers the state of the art in NK-CAR therapy as an emerging and promising therapeutic option for AML while keeping a balanced view between the advantages and disadvantages of this therapeutic approach. The authors review this subject in enough depth that is suitable for readers unfamiliar with the field but still compelling for experts in NK cell ACT.

Below are minor suggestions that can however be useful to improve the overall value of the review:

1 – Table 1: whenever possible insert appropriate references that will support the rankings herein presented. Furthermore, the authors can include in the comparison section with CAR-T cell, a brief entry of both viral and non-viral CAR gene delivery methods in NK cells. Despite it not being the direct focus of the review, this remains a relevant, often considered limitation, of engineering primary NK cells when compared to T cells.

2 – Line 102: 2017 for the FDA approval, 2018 for EMA approval?

3 – Line 111: Please consider revising the reference for the following passage: “Investigational CAR-T therapies may solve some of the limitations of established products. Targeted genome editing may allow for safe allogeneic CAR-T cells simplifying the chain of manufacture, while dual targeting and modifications to CAR-T composition could improve efficacy while reducing CRS and ICANs [25].” The referenced work from Sommermeryer et al. despite very relevant, does not seem to match this text section.

4 – Line 118: Could benefit from one comprehensive, or potentially multiple citations. Perhaps the work from Haubner et al. Leukemia 2019 (Ref44) can be useful here?: “On-target off-tumor effects of CD19 CAR-T cell therapy are limited to normal B-cells and the resulting hypogammaglobulinemia is manageable. The foremost reason that CAR T-cell therapies have not been readily adapted to AML is the absence of a similarly suitable target antigen. Many candidate AML antigens are widely expressed among myeloid cells and off-tumor effects on normal myelopoiesis can be profound. Targeting antigens present on committed myeloid precursors and mature myeloid cells leads to myelosuppression with inherent risks of infection and a requirement for advanced supportive care. If antigens expressed on hematopoietic stem cells (HSC) are targeted, marrow aplasia results, requiring stem cell transplantation for marrow recovery.”

5 – Section 2.2 Target Antigens in AML: May be become further comprehensive if a small paragraph describing recent platforms for multiple and modular platforms, even if only with pre-clinical evidence, are included as future approaches to improve multiple antigen targeting. Examples of such in the CAR setting would be the Cho et al. Cell 2018 and a similar platform in the AML setting would be the Benmebarek et al. Leukeamia 2021. Furthermore, here authors can mention that what is achieved by engineering T cells to express specific modular receptors can be done in NK cells by using ADCC inducing antibodies to one or more AML relevant antigens.

Figure 2: should be referenced in the text. I felt a relatively big conceptional jump from finishing reading section 3.2 and then looking in the figure to the right hand side and reading “Restored NK cell function; AML in remission”. I certainly believe in the potential of NK cell based therapies, but at this point of the manuscript the references and work that will support this claim have not yet been introduced – they are indeed very well described from section 3.3 on. Perhaps this figure can benefit from being partitioned into 2 separate figures? A first one showing the central concept of hallmarks of NK cell suppression and a second figure showcasing notable approaches that are able, either at pre-clinical or at clinal level, to effectively lead to AML remission?

Section 4.1 – Reference 17 should be included here? The authors mention Liu et al though.

Lines 342-343: “The presence of alloreactive NK cells in haploidentical grafts has been shown to robustly correlate with disease relapse risk” à The sentence will imply the opposite of what I think the authors intend, the word “reduced” before disease risk should be used.

Lines 386: “the group substituted exogenous  rhIL-15 à “with” should be added before substitute to clarify that IL-2 was substituted by IL-15 in the referenced clinical trial to prevent unwanted Treg expansion.

Line 695: “though TRAIL is thought to play a limited role in serial killing due to a limited reservoir of preformed ligand”à This sentence should be clarified: (Prager et al., 2019) described in an interesting work that NK cells start to rely more on the death receptor pathway during serial killing as a result of cytotoxic granules’ depletion. More so, the approach mentioned by the authors of engineering NK cells with constitutive expression of TRAIL to compensate for its shedding remains relevant. However, it should be mentioned that this approach could induce fratricide of neighboring NK or CTLs that themselves can express death receptors (Li et al., 2002).

Author Response

Comment1:

The review is very well written and covers the state of the art in NK-CAR therapy as an emerging and promising therapeutic option for AML while keeping a balanced view between the advantages and disadvantages of this therapeutic approach. The authors review this subject in enough depth that is suitable for readers unfamiliar with the field but still compelling for experts in NK cell ACT.

Below are minor suggestions that can however be useful to improve the overall value of the review:

Response 1:

We appreciate the comments and suggestions of the reviewer.

Comment 2: re Table 1- whenever possible insert appropriate references that will support the rankings herein presented. Furthermore, the authors can include in the comparison section with CAR-T cell, a brief entry of both viral and non-viral CAR gene delivery methods in NK cells. Despite it not being the direct focus of the review, this remains a relevant, often considered limitation, of engineering primary NK cells when compared to T cells.

Response 2: We appreciate the reviewer’s suggested improvements to Table 1. We have now, where feasible, referenced many of the entries in the table which compares CAR-NK with autologous and allogeneic CAR-T. Where a clear and appropriate reference was not available for associated costs of care, we have substituted ‘variable’ instead of numerical values. In addition, we agree that the compatibility of non-viral gene delivery approaches is highly relevant here. We have added an additional row to the table to address this point. To ensure that less familiar readers are introduced to this concept in the text, we have added sentences to sections 2.1 and 4.1 referencing non-viral gene delivery (where CAR-T and CAR-NK principles are first explored, respectively).

Comment 3: Line 102- 2017 for the FDA approval, 2018 for EMA approval?

Response 3: Thank you for pointing out this issue which is of interest to readers of this section introducing CAR-T cell therapies – we have amended the passage to convey the accurate information on the timing of US and EU approval.

Comment 4: Line 111- Please consider revising the reference for the following passage: “Investigational CAR-T therapies may solve some of the limitations of established products. Targeted genome editing may allow for safe allogeneic CAR-T cells simplifying the chain of manufacture, while dual targeting and modifications to CAR-T composition could improve efficacy while reducing CRS and ICANs [25].” The referenced work from Sommermeryer et al. despite very relevant, does not seem to match this text section.

Response 4: We agree with the reviewer on this point that the reference provided does not adequately reflect the points being made. We have removed this reference here and instead provide specific references describing the first in human application of CRISPR/Cas9 allogeneic CAR-T, phase I clinical data of a dual CD19/CD22 CAR-T product, clinical data of Lisocabtagene Maraleucel (with its defined CD4 and CD8 ratio, and relatively low rates of grade 3 CRS and ICANS), and finally preclinical evidence of enhanced CAR-T persistence with the addition of PI3K inhibition during manufacture [1–4]. We think this selection provides a more accurate picture of next generation approaches, which was intended with this passage.

Comment 5: Line 118- Could benefit from one comprehensive, or potentially multiple citations. Perhaps the work from Haubner et al. Leukemia 2019 (Ref44) can be useful here?: “On-target off-tumor effects of CD19 CAR-T cell therapy are limited to normal B-cells and the resulting hypogammaglobulinemia is manageable. The foremost reason that CAR T-cell therapies have not been readily adapted to AML is the absence of a similarly suitable target antigen. Many candidate AML antigens are widely expressed among myeloid cells and off-tumor effects on normal myelopoiesis can be profound. Targeting antigens present on committed myeloid precursors and mature myeloid cells leads to myelosuppression with inherent risks of infection and a requirement for advanced supportive care. If antigens expressed on hematopoietic stem cells (HSC) are targeted, marrow aplasia results, requiring stem cell transplantation for marrow recovery.”

Response 5: Again, we appreciate the reviewer flagging the need for expanded references in this section, and agree with this assessment. We have included the suggested reference, along with a further reference based on transcriptomic and proteomic data (Perna et al.) [5,6]. These references establish the absence of a single ‘ideal’ target antigen in AML. We have also added a reference providing an example of myeloablation after therapy with a dual CD33/CLL-1 CAR-T product to support the points later in the passage [7].

Comment 6: Section 2.2 Target Antigens in AML: May be become further comprehensive if a small paragraph describing recent platforms for multiple and modular platforms, even if only with pre-clinical evidence, are included as future approaches to improve multiple antigen targeting. Examples of such in the CAR setting would be the Cho et al. Cell 2018 and a similar platform in the AML setting would be the Benmebarek et al. Leukeamia 2021. Furthermore, here authors can mention that what is achieved by engineering T cells to express specific modular receptors can be done in NK cells by using ADCC inducing antibodies to one or more AML relevant antigens.

Response 6: We greatly appreciate the reviewer’s expertise on this point. We have included a passage on the applicability of modular platforms leading on from the initial discussion of prospectively identified dual targets in section 2.2 and included the suggested references. We agree that these important developments warrant discussion here. We have also acknowledged that similar principles could be achieved with CAR-NK cells in combination with monoclonal antibodies and ADCC, but to ensure the flow of the article is maintained, we make this point in the introduction to CAR-NK in section 4.

Comment 7: re Figure 2- should be referenced in the text. I felt a relatively big conceptional jump from finishing reading section 3.2 and then looking in the figure to the right hand side and reading “Restored NK cell function; AML in remission”. I certainly believe in the potential of NK cell based therapies, but at this point of the manuscript the references and work that will support this claim have not yet been introduced – they are indeed very well described from section 3.3 on. Perhaps this figure can benefit from being partitioned into 2 separate figures? A first one showing the central concept of hallmarks of NK cell suppression and a second figure showcasing notable approaches that are able, either at pre-clinical or at clinal level, to effectively lead to AML remission?

Response 7: We appreciate the reviewer’s feedback on Figure 2 and have considered it carefully. We have ensured that Figure 2 is referenced in the text. We agree that the lower right-hand portion which aimed to capture the concept of restoration of NK function in remission represents a conceptual jump when presented at this point in the manuscript. On balance we decided to simplify Figure 2 by removing this section, and instead focus solely upon the mechanisms of NK cell suppression. The concepts of restored NK function and remission maintenance across standard therapy, ASCT and NK ACT are discussed within the body of the manuscript.

Comment 8: Section 4.1 – Reference 17 should be included here? The authors mention Liu et al though.

Response 8: We appreciate that this omission was pointed out and have included the reference here in the revised manuscript.

Comment 9: Lines 342-343: “The presence of alloreactive NK cells in haploidentical grafts has been shown to robustly correlate with disease relapse risk” à The sentence will imply the opposite of what I think the authors intend, the word “reduced” before disease risk should be used.

Response 9: We agree with the reviewer that this is misleading and have amended the passage to read ‘reduced disease relapse risk’.

Comment 10: re Lines 386- “the group substituted exogenous rhIL-15 à “with” should be added before substitute to clarify that IL-2 was substituted by IL-15 in the referenced clinical trial to prevent unwanted Treg expansion.

Response 10: Again we agree with the reviewer that the wording of this passage is unclear. We have amended the sentence to read ‘utilized rhIL-15 in place of IL-2’.

Comment 11: re Line 695: “though TRAIL is thought to play a limited role in serial killing due to a limited reservoir of preformed ligand”à This sentence should be clarified: (Prager et al., 2019) described in an interesting work that NK cells start to rely more on the death receptor pathway during serial killing as a result of cytotoxic granules’ depletion. More so, the approach mentioned by the authors of engineering NK cells with constitutive expression of TRAIL to compensate for its shedding remains relevant. However, it should be mentioned that this approach could induce fratricide of neighbouring NK or CTLs that themselves can express death receptors (Li et al., 2002).

Response 11: We appreciate these comments and agree with the points raised. We have taken the opportunity to review the paper by Prager et al. and have revised the passage to acknowledge this evidence suggesting the importance of death receptor mediated killing to NK serial killing, and the limitation which was also observed to repeated death receptor mediated killing which we believe clarifies the point. In addition, we have included the suggested reference (Li et al), alongside other references which support a need for caution in modulating death receptor pathways [8–10].

References:

  1. Stadtmauer, E.A.; Fraietta, J.A.; Davis, M.M.; Cohen, A.D.; Weber, K.L.; Lancaster, E.; Mangan, P.A.; Kulikovskaya, I.; Gupta, M.; Chen, F.; et al. CRISPR-engineered T cells in patients with refractory cancer. Science (80-. ). 2020, 367, doi:10.1126/science.aba7365.
  2. Hu, Y.; Zhang, Y.; Zhao, H.; Wang, Y.; Nagler, A.; Chang, A.H.; Huang, H. CD19/CD22 Dual-Targeted Chimeric Antigen Receptor T-Cell Therapy for Relapsed/Refractory Aggressive B-Cell Lymphoma: a Safety and Efficacy Study. Blood 2020, 136, 34–34, doi:10.1182/blood-2020-143239.
  3. Abramson, J.S.; Palomba, M.L.; Gordon, L.I.; Lunning, M.A.; Wang, M.; Arnason, J.; Mehta, A.; Purev, E.; Maloney, D.G.; Andreadis, C.; et al. Lisocabtagene maraleucel for patients with relapsed or refractory large B-cell lymphomas (TRANSCEND NHL 001): a multicentre seamless design study. Lancet 2020, 396, 839–852, doi:10.1016/S0140-6736(20)31366-0.
  4. Zheng, W.; O’Hear, C.E.; Alli, R.; Basham, J.H.; Abdelsamed, H.A.; Palmer, L.E.; Jones, L.L.; Youngblood, B.; Geiger, T.L. PI3K orchestration of the in vivo persistence of chimeric antigen receptor-modified T cells. Leukemia 2018, 32, 1157–1167, doi:10.1038/s41375-017-0008-6.
  5. Haubner, S.; Perna, F.; Köhnke, T.; Schmidt, C.; Berman, S.; Augsberger, C.; Schnorfeil, F.M.; Krupka, C.; Lichtenegger, F.S.; Liu, X.; et al. Coexpression profile of leukemic stem cell markers for combinatorial targeted therapy in AML. Leukemia 2019, 33, 64–74, doi:10.1038/s41375-018-0180-3.
  6. Perna, F.; Berman, S.H.; Soni, R.K.; Mansilla-Soto, J.; Eyquem, J.; Hamieh, M.; Hendrickson, R.C.; Brennan, C.W.; Sadelain, M. Integrating Proteomics and Transcriptomics for Systematic Combinatorial Chimeric Antigen Receptor Therapy of AML. Cancer Cell 2017, 32, 506-519.e5, doi:10.1016/j.ccell.2017.09.004.
  7. Liu, F.; Cao, Y.; Pinz, K.; Ma, Y.; Wada, M.; Chen, K.; Ma, G.; Shen, J.; Tse, C.O.; Su, Y.; et al. First-in-Human CLL1-CD33 Compound CAR T Cell Therapy Induces Complete Remission in Patients with Refractory Acute Myeloid Leukemia: Update on Phase 1 Clinical Trial. Blood 2018, 132, 901–901, doi:10.1182/blood-2018-99-110579.
  8. Iyori, M.; Zhang, T.; Pantel, H.; Gagne, B.A.; Sentman, C.L. TRAIL/DR5 Plays a Critical Role in NK Cell-Mediated Negative Regulation of Dendritic Cell Cross-Priming of T Cells. J. Immunol. 2011, 187, 3087–3095, doi:10.4049/jimmunol.1003879.
  9. Li, J.H.; Rosen, D.; Sondel, P.; Berke, G. Immune privilege and FasL: Two ways to inactivate effector cytotoxic T lymphocytes by FasL-expressing cells. Immunology 2002, 105, 267–277, doi:10.1046/j.1365-2567.2002.01380.x.
  10. Sag, D.; Ayyildiz, Z.O.; Gunalp, S.; Wingender, G. The role of trail/drs in the modulation of immune cells and responses. Cancers (Basel). 2019, 11, doi:10.3390/cancers11101469.

Reviewer 3 Report

This is a well-written, detailed, up-to-date, and comprehensive review.

A discussion of the advantages and challenges of hiPSC-derived off-the-shelf CAR-NK cell products would add value to the review.

In conventional CAR-T therapy, the ratio of  different types of T cells (e.g., CD4+  to D8+) appears to affect efficacy. I would like to see is a discussion on subpopulations of NK cells (eg. CD56dim vs CD56high NK cells) and how that may potentially affect the efficacy of CAR-NK cells. 

A significant limitation of conventional CAR-T based therapies is that leukemic cells can escape if they stop expressing CD19 on the surface. CAR-NK cell based therapies can be engineered to respond to two targets using a two separate mechanisms (CAR and antibody-based). Targeting 2 tumor-associated antigens may indeed be easier to accomplish with NK cells than with T cells. This concept could be explored in more depth. For example, the authors do briefly discuss the Fate product, but they do not discuss the multi-antigen targeting design of that product.

Author Response

Comment 1: This is a well-written, detailed, up-to-date, and comprehensive review.

Response 1: We appreciate the comments of the reviewer.

Comment 2: A discussion of the advantages and challenges of hiPSC-derived off-the-shelf CAR-NK cell products would add value to the review.

Response 2: We appreciate the reviewer’s feedback on this point and agree with their assessment. We have expanded the paragraph in Section 4.1, introducing the principles of CAR-NK cells, to discuss the merits of the iPSC platform in greater detail.

Comment 3: In conventional CAR-T therapy, the ratio of different types of T cells (e.g., CD4+  to D8+) appears to affect efficacy. I would like to see is a discussion on subpopulations of NK cells (eg. CD56dim vs CD56high NK cells) and how that may potentially affect the efficacy of CAR-NK cells. 

Response 3: We thank the reviewer for this suggestion and agree that this is an interesting point worth expanding upon. In section 4.1, we have expanded our discussion of the paper by Oei et al. which demonstrated the impact of innate NK cell functional potential (maturity, education and inhibitory receptors) on CAR-NK activation – analogous to the impact of T cell composition on CAR-T products [1]. This is an area for which growing clinical experience with CAR-NK cells will further refine, however we felt that the examples included best represent the potential impact of NK cell diversity.

Comment 4: A significant limitation of conventional CAR-T based therapies is that leukemic cells can escape if they stop expressing CD19 on the surface. CAR-NK cell based therapies can be engineered to respond to two targets using a two separate mechanisms (CAR and antibody-based). Targeting 2 tumor-associated antigens may indeed be easier to accomplish with NK cells than with T cells. This concept could be explored in more depth. For example, the authors do briefly discuss the Fate product, but they do not discuss the multi-antigen targeting design of that product

Response 4: We appreciate the reviewers suggestion on the topic of dual targeting. We have made several changes which help address this concern, in response to earlier comments. We have included a new section discussing the potential of modular CAR designs and the applicability to dual targeting. Furthermore, we have emphasized in section 4.1 (the introduction to CAR-NK cells) the potential for multi-antigen targeting via CD16 expression. As an example of this, we have now further described the multi-antigen approach of FT596 (Fate Therapeutics multi-antigen targeted product).

Reference:

1. Oei, V.Y.S.; Siernicka, M.; Graczyk-Jarzynka, A.; Hoel, H.J.; Yang, W.; Palacios, D.; Sbak, H.A.; Bajor, M.; Clement, D.; Brandt, L.; et al. Intrinsic functional potential of NK-Cell subsets constrains retargeting driven by chimeric antigen receptors. Cancer Immunol. Res. 2018, 6, 467–480, doi:10.1158/2326-6066.CIR-17-0207